# Improving Flavonoid Accumulation of Bioreactor-Cultured Adventitious Roots in *Oplopanax elatus* Using Yeast Extract

**DOI:** 10.3390/plants12112174

**Published:** 2023-05-30

**Authors:** Mei-Yu Jin, Miao Wang, Xiao-Han Wu, Ming-Zhi Fan, Han-Xi Li, Yu-Qing Guo, Jun Jiang, Cheng-Ri Yin, Mei-Lan Lian

**Affiliations:** 1Department of Chemistry, Yanbian University, Park Road 977, Yanji 133002, China; jinmeiyu@ybu.edu.cn; 2Agricultural College, Yanbian University, Park Road 977, Yanji 133002, China; 17704338616@163.com (M.W.); 18651676603@163.com (X.-H.W.); fan6252019@163.com (M.-Z.F.); 13614351819@163.com (H.-X.L.); 15204849165@163.com (Y.-Q.G.); jiangjun@ybu.edu.cn (J.J.)

**Keywords:** *Oplopanax elatus*, yeast extract, elicitor concentration, elicitation time, elicitation duration, flash extraction

## Abstract

*Oplopanax elatus* is an endangered medicinal plant, and adventitious root (AR) culture is an effective way to obtain its raw materials. Yeast extract (YE) is a lower-price elicitor and can efficiently promote metabolite synthesis. In this study, the bioreactor-cultured *O. elatus* ARs were treated with YE in a suspension culture system to investigate the elicitation effect of YE on flavonoid accumulation, serving for further industrial production. Among YE concentrations (25−250 mg/L), 100 mg/L YE was the most suitable for increasing the flavonoid accumulation. The ARs with various ages (35-, 40-, and 45-day-old) responded differently to YE stimulation, where the highest flavonoid accumulation was found when 35-day-old ARs were treated with 100 mg/L YE. After YE treatment, the flavonoid content increased, peaked at 4 days, and then decreased. By comparison, the flavonoid content and antioxidant activities in the YE group were obviously higher than those in the control. Subsequently, the flavonoids of ARs were extracted by flash extraction, where the optimized extraction process was: 63% ethanol, 69 s of extraction time, and a 57 mL/g liquid–material ratio. The findings provide a reference for the further industrial production of flavonoid-enriched *O. elatus* ARs, and the cultured ARs have potential application for the future production of products.

## 1. Introduction

*Oplopanax elatus* Nakai is a perennial shrubby plant of the Araliaceae family, mainly distributed in Changbai Mountain area, with some distribution in Russia and Korea [1]. *O. elatus* has high pharmacological effects, such as antioxidant, anti-inflammatory, anticancer, and antidepression [1]. However, the exploitation and utilization of *O. elatus* are delayed due to the shortage of raw materials caused by an uncontrollable collection of wild resources and the immature technology of artificial cultivation [2]. Adventitious roots (ARs) of plants can be mass and rapidly produced using plant tissue culture technology [2], and AR culture is an efficient way to obtain raw materials of rare medicinal plants [3]. Recently, *O. elatus* ARs have been successfully cultured in batch [3,4,5] or fed-batch bioreactor culture systems [2,6], and the fed-batch culture shows a high efficiency and has promising application value [6]. In addition, *O. elatus* AR cultures contain various useful metabolites, and flavonoids, as one of the main bioactive compounds, have important biological effects, especially antioxidant activity [5]. Therefore, the production of *O. elatus* flavonoids using AR culture could contribute to the development of antioxidant products in the future.

Bioreactor culture technology has been extensively applied in plant AR culture [2,6,7,8,9,10], and the ARs of various plant species have been produced in large-scale bioreactor systems [11,12,13]. In AR bioreactor culture of medicinal plants, metabolite synthesis can be controlled by various strategies, such as improving the culture medium and air volume or inoculation density, and implementing elicitation, among which the elicitation strategy is the most efficient for enhancing metabolite synthesis [14]. For example, Ho et al. [15] indicated that 50 μM methyl jasmonate (MeJA) greatly improves the phenolic and flavonoid accumulation of *Polygonum multiflorum* ARs. Le et al. [16] demonstrated that the saponin content obviously increases when *Panax ginseng* ARs are treated with a biotic elicitor (*Mesorhizobium amorphae*, GS3037). For *O. elatus* AR culture, we previously treated *O. elatus* ARs with MeJA (200 μM) and found that flavonoid and phenolic contents were considerably enhanced in the batch culture system [3]. However, the increased cost of AR culture caused by the high price of elicitors, such as MeJA, must be considered in large-scale industrial production. Consequently, excavating effective and economic elicitors is crucial for the production of useful metabolites in a culture system. An elicitor, as a trigger factor, can promote metabolite synthesis by regulating several enzyme activities; the elicitation effects and their underlying mechanisms vary among the different elicitors [14]. Yeast extract (YE) is a natural product obtained from yeast by separation and purification, and contains a variety of components, such as polypeptide, amino acid, flavor nucleotide, trace elements, and B vitamins [17]. The price of YE is much cheaper compared with MeJA, but its enhancement effect on bioactive compound accumulation in plant cell/organ culture has been repeatedly mentioned [18,19]. For instance, Goncharuk et al. [18] elucidated that YE treatment could increase phenolic compounds accumulation during *Linum grandiflorum* cell culture. Maqsood and Abdul [19] indicated that YE is beneficial for alkaloid accumulation in the hairy root culture of *Catharanthus roseus*. However, the elicitation effect of YE has not been investigated in *O. elatus* AR culture. Therefore, in this study, *O. elatus* ARs produced in a fed-batch bioreactor culture system were treated with YE in a suspension flask culture system, and the suitable YE concentration, AR age contacting YE, and YE treatment duration were selected for the enhancement of flavonoid accumulation. Then, the elicitation effect of YE was verified in the fed-batch bioreactor culture system by comparing flavonoid contents and antioxidant activities of YE-treated and -untreated ARs. This study aimed to confirm a suitable YE treatment method for obtaining large amounts of flavonoids through *O. elatus* AR culture and to provide a reference for the production of antioxidant products.

The optimization of the extraction process is an important step before the production of products. Flash extraction was firstly proposed in 1993 by Liu et al. [20], who indicated that flash extraction has many advantages, such as sufficient extraction and saving time, solvent, and energy. In this study, the flavonoids of YE-treated fed-batch bioreactor-cultured ARs were extracted by flash extraction, and the extraction process was optimized using the response surface method (RSM) to provide a reference for the further utilization of *O. elatus* ARs.

## 2. Results

### 2.1. Effect of YE Concentration on Flavonoid Accumulation of Fed-Batch Bioreactor-Cultured ARs

After 45 days of fed-batch bioreactor culture, ARs were treated with different concentrations of YE for 8 days, and then AR dry weight (DW) and flavonoid content were determined to select a suitable YE concentration. Figure 1a shows that the AR biomass did not change after YE treatment with different concentrations (25−150 mg/L). However, the total flavonoid content increased significantly (*p* < 0.05) at 75 mg/L YE and 100 mg/L YE, and the highest total flavonoid content was determined at 100 mg/L YE (Figure 1b). YE lower than 75 mg/L or higher than 100 mg/L did not exhibit an enhancement effect on flavonoid accumulation (Figure 1b). At the optimal YE concentration (100 mg/L), the total flavonoid productivity exceeded 3 g/L, which was significantly (*p* < 0.05) higher than that of the control and other YE concentration groups (Figure 1c). 

### 2.2. Effect of YE on Flavonoid Accumulation of Fed-Batch Bioreactor-Cultured ARs at Different Ages

The fed-batch bioreactor-cultured ARs at different ages (35-, 40-, and 45-day-old) were treated with 100 mg/L YE (selected in the above YE concentration experiment) for 8 days, and the elicitation effect of YE on flavonoid accumulation was investigated. AR DW did not show a significant difference among the different groups (Figure 2a). Compared with the YE-untreated control group, the total flavonoid contents in all YE groups increased significantly (*p* < 0.05) (Figure 2b). The highest total flavonoid content was found when the 30-day-old ARs were treated with YE (100 mg/L), which was approximately 70 mg/g DW more than that in the other YE treatment groups (Figure 2b). In the optimal YE treatment time, i.e., treating 35-day-old ARs with 100 mg/L YE for 8 days, the flavonoid productivity reached 4.7 g/L (Figure 2c), which was significantly higher than other YE groups (*p* < 0.05) and the control (*p* < 0.01).

### 2.3. Effect of YE Treatment Duration on Flavonoid Accumulation of Fed-Batch Bioreactor-Cultured ARs

The 35-day-old fed-batch bioreactor-cultured ARs were treated with YE (100 mg/L) on different days to investigate the elicitation effect of YE treatment duration. Figure 3a shows that AR DW did not change from 1 day to 10 days in the control and YE treatment groups. The flavonoid accumulation exerted an increase trend from 1 day to 4 days after YE treatment, peaked at 4 days (354 mg/g DW), and then decreased in the following days (Figure 3b). Compared with the control group, the total flavonoid content in the YE group was significantly (*p* < 0.01) increased on all treatment days. After 4 days of YE treatment, the flavonoid productivity reached 5.4 g/L (Figure 3c), which was 1.5 g/L higher than that of the control. 

### 2.4. Comparison of Flavonoid Contents and Antioxidant Activities of YE-Treated and YE-Untreated ARs in Fed-Batch Bioreactor Culture System

The ARs were cultured in the fed-batch bioreactor (5 L) system and treated with 100 mg/L YE for 4 days on day 35 according to the result of the above YE elicitation experiments, and the contents of total flavonoids and rutin, quercetin, and kaempferide were compared with those in the control. Table 1 shows that, in the YE group, the contents of total flavonoids (582.2 mg/g DW), rutin (2.9 mg/g DW), and kaempferide (161.9 μg/g DW) were significantly (*p* < 0.05) higher than those in the control; however, the quercetin content was not significantly (*p* < 0.05) affected by the YE treatment. The AR DW in the YE (67.17 ± 5.6 g/bioreactor) and control (65.14 ± 7.1 g/bioreactor) groups did not show a significant (*p* < 0.05) difference. The total flavonoid productivity in the YE treatment group reached 7.8 g/L, whereas that in the control group was only 4.6 g/L.

Subsequently, antioxidant activities of extracts from YE-treated (YE-ARE) and YE-untreated ARs (C-ARE) were compared by evaluating abilities of scavenging 2,2-diphenyl-1-picrylhydrazyl (DPPH) and 2′2′-azino-bis(3-ethylbenzothiazoline-6-sulfonic acid) (ABTS)^+^ radicals and chelating Fe^2+^. As shown in Figure 4, when YE-ARE concentrations increased from 15.6 μg/mL to 31.3 μg/mL, the DPPH scavenging rate dramatically increased, and then gently rose until 500 μg/mL YE-ARE; the DPPH scavenging rate in the control group increased when increasing the C-ARE concentrations (Figure 4a). At all extract concentrations, DPPH scavenging rates in the YE-ARE group were higher than those in the C-ARE group. The concentration for 50% of the maximal effect (EC_50_) in the YE-ARE group was only 19.8 μg/mL, whereas that in the C-ARE was 27.6 μg/mL. The ABTS^+^ scavenging rate (Figure 4b) and Fe^2+^ chelating rate (Figure 4c) in the YE-ARE and C-ARE groups exerted a similar pattern, which increased when the AR extract concentration increased. The EC_50_ values in the YE-ARE group regarding ABTS^+^ scavenging (0.85 mg/mL) and Fe^2+^ chelating rates (4.61 mg/mL) were lower than those in the C-ARE group. The results of DPPH and ABTS^+^ scavenging and Fe^2+^ chelating rates indicated the high antioxidant activity of YE-treated ARs, which may be related to the high contents of flavonoids contained in the YE-treated ARs (Table 1). The findings elucidate that YE-treated *O. elatus* ARs have potential value for use in the development of antioxidant products.

### 2.5. Flash Extraction Process Optimization of Flavonoids from YE-Treated Fed-Batch Bioreactor-Cultured ARs

To future utilize the *O. elatus* AR cultures, this study used the flash extraction method to extract the flavonoids of YE-treated ARs from the fed-batch bioreactor culture, and the extraction process was optimized by RSM. To provide a basis for the RSM experiment, single-factor experiments were implemented. Figure 5 shows that flavonoid yields were significantly different among the three solvent groups (water, ethanol, and methanol), and the highest value was found when ethanol was used as the solvent (Figure 5a). Then, 60% ethanol, 70 s of extraction time, and a 60 mL/g liquid−material ratio were selected through conducting the experiments of ethanol concentration (Figure 5b), extraction time (Figure 5c), and liquid−material ratio (Figure 5d). The single experiment results were used in the following RSM experiment.

The RSM experiment was designed based on a Box–Behnken design with three factors (ethanol concentration [*X*_1_], extract time [*X*_2_], and liquid−material ratio [*X*_3_]) at three levels. Each independent variable was coded at three levels between −1, 0, and +1 corresponding to the low, mid, and high level (Table 2), which included a total of 17 combination groups (Table 3). ARs were extracted depending on the extraction conditions of each combination group, and the extraction efficiency was evaluated with the flavonoid yield as the index. Table 3 shows that the flavonoid yield considerably differed among the groups, where high flavonoid yields (>7%) occurred at the central points (group 2, 5, 16, and 17). 

Further, the experimental data were fitted with various models (linear model, interactive model, quadratic model, and cubic model), and the regression equation was obtained, which was Y (flavonoid yield) = 7.06 + 0.3*X*_1_ + 0.096*X*_2_ − 0.17*X*_3_ − 0.23*X*_1_*X*_2_ − 0.37*X*_1_*X*_3_ + 0.36*X*_2_*X*_3_ − 0.78*X*_1_^2^ − 0.61*X*_2_^2^ − 0.52*X*_3_^2^. The predicted flavonoid yield in each group was calculated according to the regression equation, showing that it did not deviate much from the actual data of the experiment (Table 3).

The result of the model adequacy test shows that the lack of fit (0.2712) was not significant and that the correlation coefficient (*R*^2^) was high (0.9487) (Table 4), indicating that the model was valid [21]. Ethanol concentrations significantly (*p* < 0.01) affected the flavonoid yield, whereas the extraction time or liquid−material ratio did not show a significant effect. The order of *F* values was: ethanol concentration (11.64) > liquid−material ratio (3.73) > extraction time (1.18), indicating that the solvent concentration had the greatest influence on the extraction efficiency, followed by the liquid−material ratio and extraction time. In addition, Table 4 also shows that the interactions between the ethanol concentration and liquid−material ratio (*p* = 0.0226) and the extraction time and liquid−material ratio (*p* = 0.0247) were significant (*p* < 0.05), whereas that between the ethanol concentration and extraction time was not significant (*p* = 0.1159).

The response surface plots also exhibited an interaction between both independent variables (Figure 6). A nearly circular contour line is found in Figure 6a, indicating a minor interaction between the ethanol concentration and extraction time; by contrast, oval contour lines are found in Figure 6b,c, indicating high interactions between the ethanol concentration and liquid−material ratio and the extraction time and liquid−material ratio [22], which is consistent with Table 4. 

The optimal ethanol concentration (63%), extraction time (69 s), and liquid−material ratio (57 mL/g) were obtained by applying the methodology of the desired function, and the predicted flavonoid yield was 7.12%. Then, the optimized extraction conditions were verified through three repeated tests. Table 5 shows that the flavonoid yields in the triplet repeated tests were 7.04, 7.14, and 7.06%, with an average value of 7.08% and small relative standard deviation (1.58), indicating that the result of the triplet repeated tests was stable and close to the predicted value (7.12%) and proving that the optimized extraction process was feasible.

## 3. Discussion

Bioreactor culture of *O. elatus* ARs has been repeatedly studied in recent years using batch and fed-batch culture methods [2,4,5,6,10]. Compared with batch culture, biomass and bioactive compounds increase dramatically in fed-batch bioreactor culture because the medium inhibition in the early culture stage can be avoided and the nutrient depletion in mid or late culture stage can be replenished. However, efforts to increase bioactive compounds need to be continued during the culture. MeJA as an elicitor has a good elicitation role in many culture systems [4,23,24,25,26], including *O. elatus* AR culture. Jiang et al. [4] indicated that MeJA (200 μM) increases bioactive compound production in the batch culture of *O. elatus* ARs. However, the use of MeJA could greatly increase the production cost. Thus, the selection of economical and efficient elicitors is vital for commercial production. In this study, YE was selected as an elicitor to investigate its improvement effect on flavonoid accumulation, aiming to apply the elicitation method on the further large-scale fed-batch bioreactor culture for the mass production of flavonoid-enriched *O. elatus* ARs. 

### 3.1. Optimization of YE Elicitation

Studies have indicated that plant metabolite synthesis can be promoted by various elicitors, including YE [27,28,29]. However, no elicitation method is suitable for all culture systems. The elicitation efficiency is affected by several factors, such as elicitor types, elicitor concentrations, elicitation treatment time, and duration [30]. Therefore, the selection of an appropriate elicitation strategy is essential for improving the production of useful metabolites. 

The concentration effect of elicitors can be divided into two types, i.e., the reaction saturation type and optimum concentration type [31]. For the reaction saturation type, secondary metabolite synthesis increases with an increase in elicitor concentrations, and remains stable after reaching the maximum value. For the optimum concentration type, secondary metabolite synthesis reaches the highest value at a certain level of elicitor concentration, but decreases when the elicitor concentration continues to increase, where most plant culture is of this type [31]. The effect of YE concentration has been investigated by several researchers. For instance, Vijayalakshmi and Shourie [29] indicated that, among the tested YE concentrations (25−175 mg/L), treating with 75 mg/L YE for 10 days is the most suitable for promoting the flavonoid accumulation of *Glycyrrhiza glabra* calluses, where the flavonoid yield is increased more than two-fold. Kochan et al. [32] obtained the maximum yield of ginsenosides (Rb1, Rb2, Rc, Rd, Re, and Rg1) in hairy roots of *Panax quinquefolium* at 50 mg/mL YE, which is obviously higher than that of the other tested YE concentrations (50−2000 mg/L). In this study, we found that the total flavonoid content of *O. elatus* ARs increased when increasing the YE concentrations, peaked at 100 mg/L YE, and then decreased when the YE concentration was higher than 100 mg/L, elucidating that the concentration effect of YE on the flavonoid synthesis of *O. elatus* ARs belongs to the optimum concentration type. These findings prove that the suitable YE concentration is not equal in different culture systems and indicates the importance of selecting elicitor concentrations. 

At different stages of growth and development, the sensitivity of plant cells to elicitors varies, and cells that accumulate a certain biomass can effectively receive the elicitor signals and show a high activity [8]. The elicitation efficiency depends on the ages of cultures, and generally shows a good effect when an elicitor stimulates plant cell/organ cultures at the exponential or stationary growth stage. For example, Loc and Giang [28] treated cell cultures of *Centella asiatica* with YE (4 g/L) on day 5, day 10, and day 15 of culture, and suggested that YE treating 10-day-old cells is the most suitable for the production of asiaticosides. In this study, compared with YE treating 40- and 45-day-old ARs, flavonoid accumulation significantly (*p* < 0.01) increased when 35-day-old *O. elatus* ARs were treated with YE (100 mg/L). 

The effect of elicitation duration is also critically important. Because of the different defense response rates of plant cells, the suitable elicitation duration differs in each culture system [8]. In plant cell/organ culture, metabolite accumulation tends to increase after a certain elicitation duration and then decrease with an extension of the duration [33]. Farjaminezhad and Garoosi [27] demonstrated that a high azadirachtin content (16.1 mg/g DW) occurs when *Azadirachta indica* cells are treated with YE (25 mg/L) for 2 days. Chen and Chen [34] indicated that an increase in cryptotanshinone is achieved after 7 days of YE treatment in *Ti*-transformed *Salvia miltiorrhiza* cell culture. In this study, we found that treating 35-day-old fed-batch cultured *O. elatus* ARs with YE (100 mg/L) for 4 days was most favorable for enhancing flavonoid synthesis.

The elicitation effect of YE in fed-batch *O. elatus* AR culture was clarified in this study. Meanwhile, our recent study found that salicylic acid could also greatly increase the bioactive compound accumulation of fed-batch cultured *O. elatus* ARs [35]. Therefore, to maximize the production of the useful bioactive compounds, the combined use of elicitors should be investigated in further studies. 

### 3.2. Flavonoid Extraction Using Flash Extraction

Flash extraction is a new extraction technology with many advantages, such as being fast and safe, saving energy, environmental protection, and having a high efficiency. Recently, flash extraction has been used to extract various primary (such as polysaccharides, oils, proteins) and secondary metabolites (such as phenols, alkaloids, and terpenoids) of plants [36]. 

In flash extraction, many factors, such as solvent types and concentrations, the extraction time, the extraction temperature, and the liquid−material ratio, affect the extraction efficiency [37]. To optimize the flash extraction process, various experimental design methods have been applied, among which RSM is commonly used [38]. For example, Zhang et al. [39] optimized the flash extraction process using RSM for the extraction of phenolics from *Eriobotrya japonica* leaves and indicated that the optimal extraction conditions are 62% ethanol as the solvent, a 32 mL/g liquid–material ratio, and 127 s of extraction time, extracting twice. Li et al. [40] extracted flavonoids from *Crotalaria ferruginea* and applied RSM to optimize the flash extraction process, and the optimal extraction process was 60% ethanol as the solvent, 92 s of extraction time, and a 35 mL/g liquid−material ratio. In this study, we extracted flavonoids of *O. elatus* ARs using flash extraction and used RSM to optimize the extraction process. The result indicates that the optimal extraction process was: 67% ethanol as the solvent, 67 s of extraction time, and a 57 mL/g liquid−material ratio. Under this optimal extraction conditions, the flavonoid yield was higher than 7%. This study proved that the flash extraction method was suitable for the extraction of *O. elatus* ARs, which provides a theoretical basis for rapid and convenient extraction in further large-scale production.

## 4. Materials and Methods

### 4.1. Plant Materials and YE Solution Preparation

*O. elatus* ARs were induced from in vitro cultured seedling roots. The induced ARs were cut into approximately 1 cm length and batch cultured in a 5 L balloon-type airlift bioreactor according to the method of Jiang et al. [5]. After 30 days of culture, the ARs were used for fed-batch bioreactor culture.

YE powder (Shanghai Yuanye Bio-Technology Co., Ltd., Shanghai, China) was dissolved and diluted with deionized water. The YE solution was sterilized through a membrane filter (0.22 μm) and then used in the following YE treatment experiments.

### 4.2. Fed-Batch AR Culture

The 20 g (fresh weight, FW) ARs were inoculated in a 5 L bioreactor with 3 L of half-strength Murashige and Skoog (MS) medium [41] supplemented with 3 mg/L indolebutyric acid (IBA) (Shanghai Yuanye Bio-Technology Co., Ltd., Shanghai, China) and 40 g/L sucrose. After 15 days of culture, 2 L feeding medium (MS + 3 mg/L IBA + 60 g/L sucrose) was added to the bioreactor. The pH of culture medium was adjusted to 5.8 prior to autoclaving. The bioreactor was aerated at 0.1 vvm (air volume/medium volume/min) and maintained at 25 °C in the dark.

### 4.3. YE Elicitation Experiment Design 

The experiment process is shown in Figure 7, namely ARs and culture medium were collected from the bioreactor of fed-batch culture and transferred to flasks to conduct YE treatment experiments; the confirmed YE treatment method was verified in the fed-batch bioreactor system.

Elicitation experiments were performed using 250 mL Erlenmeyer flasks. In the first experiment, the effect of YE concentration was investigated. ARs (20 g, FW) were inoculated in the 5 L bioreactor and feeding medium was added after 15 days of initial culture. After 45 days (total culture days) of culture, approximately 600 g fresh ARs and 3 L culture medium were collected. A total of 100 mL medium and 20 g fresh ARs was added to the flask, as well as different concentrations of YE (25, 50, 75, 100, 125, and 150 mg/L); an equal amount of the medium was added to the control group (YE = 0 mg/L). After 8 days of YE treatment, ARs were harvested from the flasks, and the AR biomass (DW) and total flavonoid content were determined. In the second experiment, the effect of YE on flavonoid accumulation of ARs at different ages was investigated. ARs (20 g, FW) were inoculated in 5 L bioreactors, and feeding medium was added after 15 days of initial culture. ARs and culture medium were separately collected from bioreactors after 35, 40, and 45 days (total culture days) of culture. A total of 20 g fresh ARs and 100 mL medium were added to the flask, and then 100 mg/L YE was added according to the result of YE concentration experiment; equal amount of medium was added to the control group. After 8 days of YE treatment, ARs were harvested from the flasks, and AR DW and total flavonoid content were determined. In the third experiment, the effect of YE treatment duration was investigated. ARs (20 g, FW) were inoculated in 5 L bioreactors, and feeding medium was added after 15 days of initial culture. ARs and culture medium were collected from bioreactors after 35 days (total culture days) of culture according to the result of above experiment. A total of 20 g fresh ARs and 100 mL medium were added to the flasks, and then 100 mg/L YE was added according to the result of YE concentration experiment; an equal amount of medium was added to the control group. The ARs were sampled at 1-day intervals, and the AR DW and total flavonoid content were determined. All flasks were kept at 100 rpm on a shaker at 25 °C in the dark.

### 4.4. Comparison of Flavonoid Contents and Antioxidant Activities between YE-Treated and YE-Untreated ARs in Fed-Batch Bioreactor Culture 

The elicitation effect of YE was verified in the fed-batch bioreactor culture system, and the flavonoid contents and antioxidant activities between YE-treated and YE-untreated ARs were compared. 

In the YE group, 100 mg/L YE was added to the bioreactor on day 35 of fed-batch bioreactor culture, and the ARs were harvested after 4 days of YE treatment (39 days of total culture period). In the control group, the ARs were harvested after 39 days of fed-batch bioreactor culture. At the end of the bioreactor culture, AR dry weight, the contents of total flavonoids, rutin, quercetin, and kaempferide, and rates of scavenging DPPH, ABTS^+^ radicals, and chelating Fe^2+^ were determined. 

### 4.5. Optimization of Extraction Process 

To extract flavonoids, the YE-treated fed-batch bioreactor-cultured *O. elatus* ARs were extracted using a flash extractor (Shanghai Precision Equipment Co., Ltd., Shanghai, China) with a voltage of 220 V and a revolution of 10,000 rpm. 

A total of 3 g dry ARs and solvent were added into an extraction vessel (500 mL) and then extracted for the schedule time. The mixture solution was passed through a sieve (38 μm) and the filtrate was collected. The filtrate was concentrated with a rotary evaporator and then lyophilized to obtain the dry extract. The dry extract weight and the flavonoid content were determined and then flavonoid yield was calculated according to the following formula and used as an evaluation index.
Flavonoid yield (%) = (extract DW (g)/(AR DW [g]) × flavonoid content (mg/g DW) × 0.1

To optimize extraction process, single-factor experiments (Table 6) were firstly conducted to select suitable solvent type and concentration, extract time, and liquid−material ratio. On the basis of the result of the single-factor experiments, the RSM was used to optimize the extraction process by adjusting ethanol (solvent) concentration, extraction time, and liquid−material ratio (Table 2). A total of 17 combination groups are shown in Table 3, and the ARs were extracted according to the relevant extraction condition of each group.

### 4.6. Determination of AR Dry Weight

The harvested ARs were washed with distilled water trice and dried at 45 °C until a constant weight was achieved after the AR surface water was removed, and then AR DW was recorded.

### 4.7. Determination of Flavonoid Content

Total flavonoid content was determined according to the method of Jin et al. [6]. In brief, the dry AR sample (0.1 g) was soaked in 10 mL of 70% ethanol, heated at 60 °C for 3 h, and filtered with a filter paper. The filtrate was used to determine the total flavonoid content via the aluminum nitrate colorimetric method at 510 nm using a spectrophotometer; rutin was used as the standard. The result was expressed as the equivalent of rutin per gram of the DW sample.

The method of Zhang et al. [42] was used to determine the contents of rutin, quercetin, and kaempferide using high-performance liquid chromatography with a C_15_ column (4.6 × 250 mm, 5 μm, Thermo Scientific, Waltham, MA, USA) and ultraviolet–visible detector (SPD-15C, Shimadzu Co., Kyoto, Japan). The mobile phases were methanol (A) and 0.1% phosphorus solution (B). Gradient elution was performed as follows: 55% B for 0−15 min and 20% B for 15−30 min. The flow rate was 0.8 mL/min. Rutin, quercetin, and kaempferide were detected at 366 nm (Figure 8). Standards of rutin (purity > 98%), quercetin (purity > 98%), and kaempferide (purity > 98%) were purchased from Shanghai Yuanye Bio-Technology Co., Ltd. (Shanghai, China).

### 4.8. Determination of Antioxidant Activities

The dry samples of YE-treated ARs were soaked in 70% ethanol and heated at 60 °C for 3 h. After filtration through a filter paper, the filtrate was lyophilized after concentrating under a rotary evaporation. The dry extract was dissolved in deionized water and the different concentrations of the AR extract were prepared using double dilution method. The antioxidant activities of AR extracts of different concentrations were evaluated by determining rates of scavenging DPPH and ATBS^+^ radicals and chelating Fe^2+^. 

The method of Jiang et al. [5] was used to determine the DPPH radical scavenging rate. In brief, a 96-well plate was added with100 μL of 10 mM DPPH (Notales Biotechnology Co., Ltd., Beijing, China) solution and 100 μL of AR extract (15.6−500 μg/mL). After 30 min of reaction in the dark, the absorbance of the mixture was determined at 517 nm using a microplate reader (iMark, Bio-Rad Laboratories, Inc., Hercules, CA, USA). To determine ABTS^+^ scavenging rate, the ABTS^+^ solution was prepared according to Fu et al. [43]. A total of 3.9 mL ABTS^+^ solution and 0.1 mL of AR extract (0.25−4 mg/mL) were mixed and reacted for 6 min in the dark. The absorbance of the mixture was determined at 734 nm (UV- T6, Beijing Purkinje General Instrument Co., Ltd., Beijing, China). The Fe^2+^ chelating rate was determined using the method of Fu et al. [43]. In brief, a 96-well plate was added with 50 μL of AR extract (1.25−20 mg/mL), 2.5 μL FeCl_2_, 10 L ferrozine, and 137.5 L deionized water, and was incubated for 10 min in the dark. The absorbance of the mixture was determined at 562 mm (iMark, Bio-Rad Laboratories, Inc., Hercules, CA, USA).

### 4.9. Statistical Analysis

All data are presented as the mean ± standard deviation of three independent replicates. Data were analyzed using Duncan’s multiple range test or Student’s *t*-test by using SPSS statistics 22.0 software (IBM Institute, Almonk, NY, USA). Values of *p* < 0.05 were considered statistically significant.

## 5. Conclusions

The YE concentration, AR age of YE treatment, and YE treatment duration critically affected the flavonoid accumulation of fed-batch bioreactor-cultured *O. elatus* ARs. The concentration effect of YE belonged to the optimum concentration type, where the optimal YE concentration was 100 mg/L; flavonoid accumulation was the most favorable when 35-day-old fed-batch cultured ARs were treated with 100 mg/L YE for 4 days, in which the total flavonoid content was 224.5 mg/g DW higher than the control; the contents of rutin and kaempferide were also greatly increased after YE treatment, whereas YE did not significantly affect the quercetin content. Flavonoid accumulation was enhanced by YE treatment, and the antioxidant activity was correspondingly increased; the rates of scavenging DPPH and ABTS and chelating Fe^2+^ in YE-ARE were higher than those in C-ARE. Flash extraction efficiently extracted flavonoids from *O. elatus* ARs, where the optimized extraction process was: 63% ethanol, 69 s of extraction time, and a 57 mL/g liquid−material ratio. The findings of this study provide a useful method for increasing the flavonoid production of *O. elatus* ARs and form a theoretical basis for the utilization of ARs in the future development of *O. elatus* ARs.

## Figures and Tables

**Figure 1 plants-12-02174-f001:**
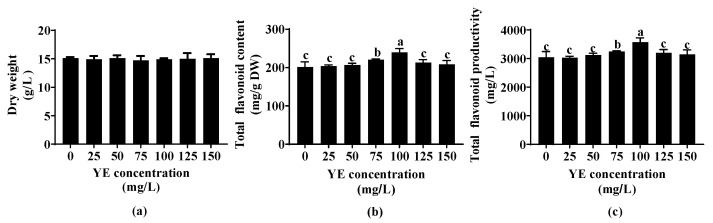
Effect of yeast extract (YE) concentration on biomass (**a**) and flavonoid content (**b**) and productivity (**c**) of *Oplopanax elatus* adventitious roots (ARs). The 45-day-old fed-batch bioreactor-cultured ARs were transferred to flasks and treated with YE for 8 days. Data represent the mean value ± standard deviation (*n* = 3). The different letters within the same column indicate significant difference by Duncan’s multiple range test at *p* < 0.05.

**Figure 2 plants-12-02174-f002:**
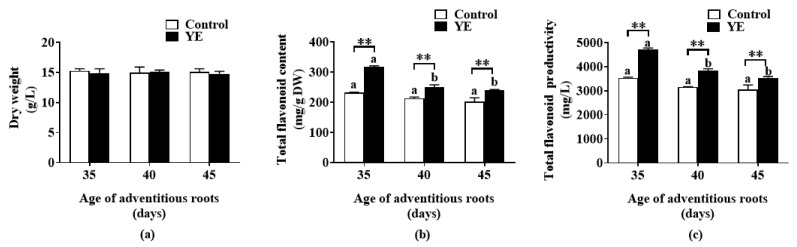
Effect of yeast extract (YE) on biomass (**a**) and flavonoid content (**b**) and productivity (**c**) of *Oplopanax elatus* adventitious roots (AR) at different ages. The 35-, 40, and 45-day-old fed-batch cultured *O. elatus* ARs were transferred to flasks and treated with 100 mg/L YE (the control group was added with equal amount of the medium) for 8 days in the YE group. Data represent the mean value ± standard deviation (*n* = 3). The different letters within the same color column indicate significant difference by Duncan’s multiple range test at *p* < 0.05. ** indicates significant difference between the groups of YE and control by Student’s *t*-test at *p* < 0.01.

**Figure 3 plants-12-02174-f003:**
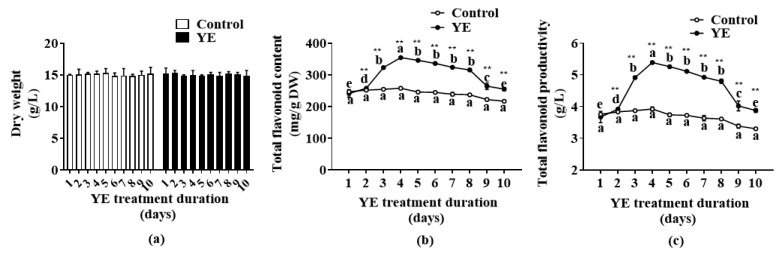
Effect of yeast extract (YE) treatment duration on biomass (**a**) and flavonoid content (**b**) and productivity (**c**) of *Oplopanax elatus* adventitious roots (ARs). The 35-day-old fed-batch bioreactor-cultured ARs were transferred to flasks and treated with 100 mg/L YE (the control group was added with equal amount of the medium) in the YE group. Data represent the mean value ± standard deviation (*n* = 3). The different letters within the same line indicate significant difference by Duncan’s multiple range test at *p* < 0.05. ** indicates significant difference between the groups of YE and control at each time point by Student’s *t*-test at *p* < 0.01.

**Figure 4 plants-12-02174-f004:**
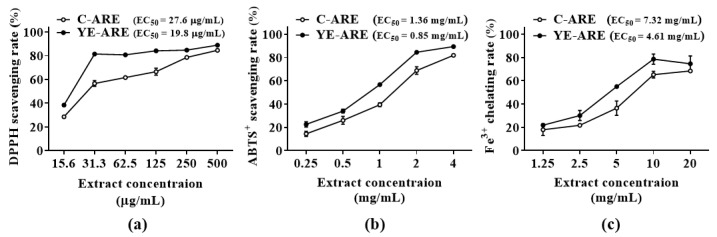
Comparison of antioxidant ability of extracts from YE-untreated (C-ARE) and -treated (YE-ARE) adventitious roots (ARs) of *Oplopanax elatus* from the fed-batch bioreactor culture. (**a**) DPPH scavenging rate. (**b**) ABTS^+^ scavenging rate. (**c**) Fe^3+^ chelating rate. The ARs in YE group were treated with 100 mg/L YE for 4 days on day 35 of fed-batch bioreactor culture. The ARs in the control group were cultured for 49 days in fed-batch bioreactor culture system. DPPH = 2,2-diphenyl-1-picrylhydrazyl; ABTS = 2′2′-azino-bis (3-ethylbenzothiazoline-6-sulfonic acid). EC_50_ = concentration for 50% of maximal effect. Data represent the mean value ± standard deviation (*n* = 3).

**Figure 5 plants-12-02174-f005:**
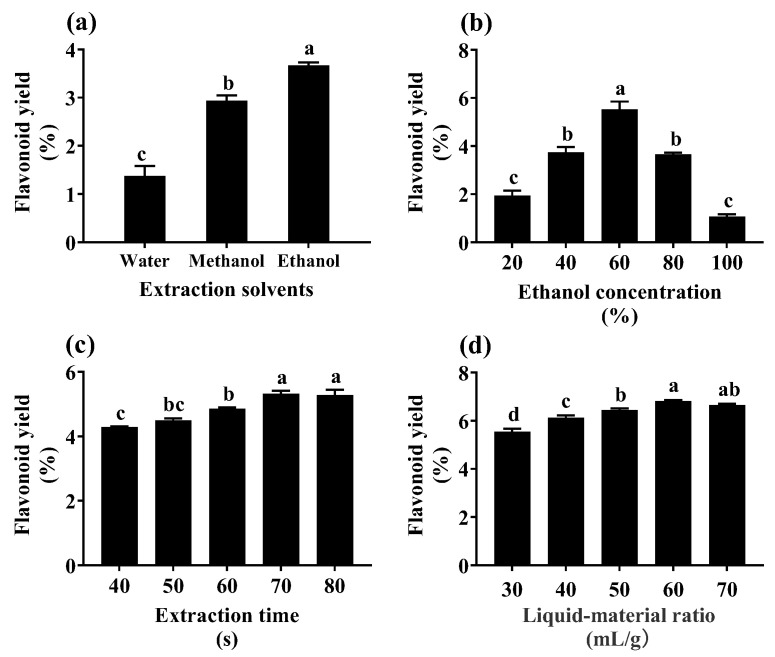
Effect of extraction solvent (**a**), solvent (ethanol) concentration (**b**), extraction time (**c**), and liquid−material ratio (**d**) on flavonoid yield of *Oplopanax elatus* adventitious roots from fed-batch bioreactor culture. Data represent the mean value ± standard deviation (*n* = 3). The different letters within the same column indicate significant difference by Duncan’s multiple range test at *p* < 0.05.

**Figure 6 plants-12-02174-f006:**
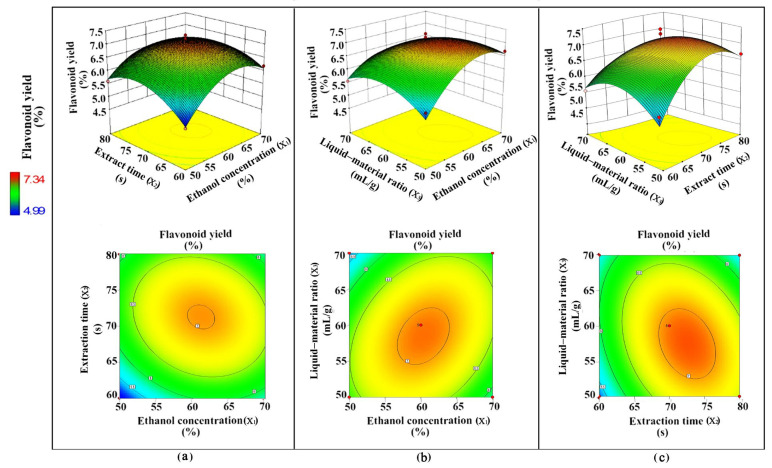
Interaction effect of two factors on flavonoid yield of extract from *Oplopanax elatus* adventitious roots (ARs). (**a**) interaction between ethanol concentration and extraction time. (**b**) interaction between ethanol concentration and liquid−material ratio. (**c**) interaction between extraction time and liquid−material ratio. The ARs were harvested from bioreactors after 4 days of yeast extract (100 mg/L) treatment on day 35 of fed-batch bioreactor culture.

**Figure 7 plants-12-02174-f007:**
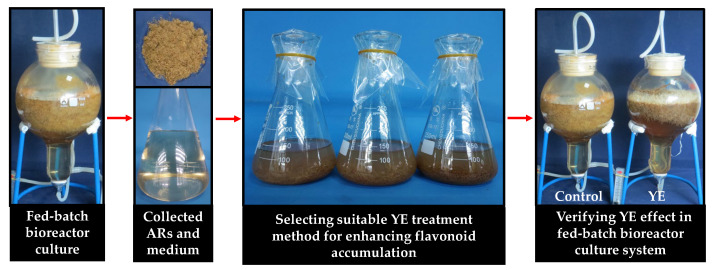
Experiment process.

**Figure 8 plants-12-02174-f008:**
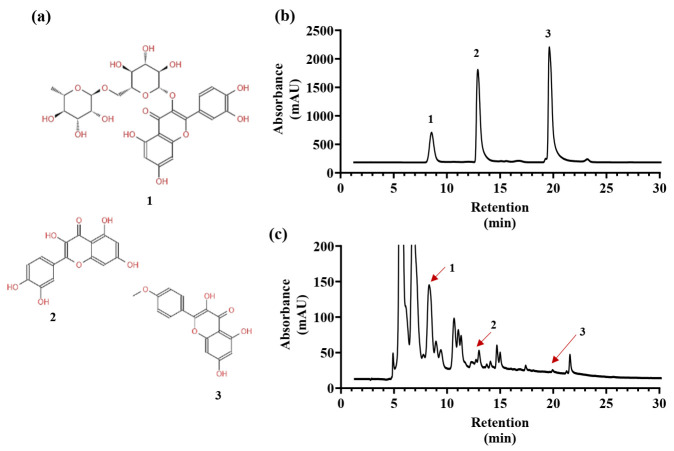
Chemical structure (**a**) and high-performance liquid chromatography profiles of rutin, quercetin, and kaempferide standards (**b**) and adventitious root sample of *Oplopanax elatus* (**c**). (1) Rutin. (2) Quercetin. (3) Kaempferide.

**Table 1 plants-12-02174-t001:** Comparison of flavonoid contents between yeast extract (YE)-treated and YE-untreated adventitious roots (ARs) of *Oplopanax elatus* in fed-batch bioreactor system.

Treatment	Total Flavonoids (mg/g DW)	Rutin (mg/g DW)	Quercetin (μg/g DW)	Kaempferide (μg/g DW)
control	357.7 ± 38.9	2.4 ± 0.1	234.5 ± 10.3	117.4 ± 7.5
YE	582.2 ± 11.7 *	2.9 ± 0.1 *	266.6 ± 11.7	161.9 ± 9.7 *

The ARs in the YE group were treated with 100 mg/L YE for 4 days on day 35 of the fed-batch bioreactor culture. The ARs in the control group were cultured for 49 days in the fed-batch bioreactor culture system. Data represent the mean value ± standard deviation (*n* = 3). * indicates significant difference between the groups of YE and control by Student’s *t*-test at *p* < 0.05.

**Table 2 plants-12-02174-t002:** Coded and experimental levels of independent variables used in response surface method experiment.

Independent Variables	Coded Level		
	−1	0	+1
ethanol concentration (*X*_1_)	50%	60%	70%
extraction time (*X*_2_)	60 s	70 s	80 s
liquid–material ratio (*X*_3_)	50 mL/g	60 mL/g	70 mL/g

**Table 3 plants-12-02174-t003:** Box–Behnken experimental design matrix and experimental responses.

Groups	Codes (Levels)			Flavonoid Yield (%)	
	Ethanol Concentration (*X*_1_)	Extraction Time (*X*_2_)	Liquid−Material Ratio (*X*_3_)	Actual Value	Predicted Value
1	0 (60%)	−1 (60 s)	1 (70 mL/g)	5.56 ± 0.12	5.66
2 ^a^	0 (60%)	0 (70 s)	0 (60 mL/g)	7.00 ± 0.03	7.06
3	−1 (50%)	−1 (60 s)	0 (60 mL/g)	4.99 ± 0.08	5.04
4	−1 (50%)	0 (70 s)	1 (70 mL/g)	5.45 ± 0.11	5.30
5 ^a^	0 (60%)	0 (70 s)	0 (60 mL/g)	6.92 ± 0.07	7.06
6	1(70%)	0 (70 s)	1 (70 mL/g)	6.39 ± 0.16	6.22
7	0 (60%)	1 (80 s)	−1 (50 mL/g)	6.69 ± 0.01	6.60
8	−1 (50%)	1 (80 s)	0 (60 mL/g)	6.19 ± 0.05	6.10
9	−1 (50%)	0 (70 s)	−1 (50 mL/g)	6.18 ± 0.15	6.36
10	1(70%)	1 (80 s)	0 (60 mL/g)	5.90 ± 0.06	5.84
11	0 (60%)	0 (70 s)	0 (60 mL/g)	6.82 ± 0.04	7.06
12	0 (60%)	−1 (60 s)	−1 (50 mL/g)	5.50 ± 0.11	5.26
13	0 (60%)	1 (80 s)	1 (70 mL/g)	5.29 ± 0.12	5.52
14	1 (70%)	−1 (60 s)	0 (60 mL/g)	5.60 ± 0.12	5.70
15	1 (70%)	0 (70 s)	−1 (50 mL/g)	5.69 ± 0.07	5.84
16 ^a^	0 (60%)	0 (70 s)	0 (60 mL/g)	7.34 ± 0.10	7.06
17 ^a^	0 (60%)	0 (70 s)	0 (60 mL/g)	7.21 ± 0.01	7.06

Data represent the mean value ± standard deviation (*n* = 3). ^a^ Central points (used to determine the experimental error).

**Table 4 plants-12-02174-t004:** Model adequacy test and ANOVA analysis.

Variables	Sun of Squares	Degree of Freedom	Mean Square	*F*	*p*
model	8.14	9	0.90	14.37	0.0010 **
*X* _1_	0.73	1	0.73	11.64	0.0113 *
*X* _2_	0.074	1	0.074	1.18	0.3137
*X* _3_	0.23	1	0.23	3.73	0.0948
*X* _1_ *X* _2_	0.20	1	0.20	3.22	0.1159
*X* _1_ *X* _3_	0.53	1	0.53	8.47	0.0226 *
*X* _2_ *X* _3_	0.51	1	0.51	8.13	0.0247 *
*X* _12_	2.55	1	2.55	40.49	0.0004 **
*X* _22_	1.57	1	1.57	24.93	0.0016 **
*X* _32_	1.14	1	1.14	18.12	0.0038 **
residual	0.44	7	0.063	-	-
lack of fit	0.26	3	0.086	1.90	0.2712
pure error	0.18	4	0.045	-	-
cor total	8.58	16	-	-	-
*R* ^2^	0.9487				

*X*_1_ = ethanol concentration. *X*_2_ = extraction time. *X*_3_ = liquid−material ratio. * and ** indicate significant differences at *p* < 0.05 and *p* < 0.01, respectively. *R*^2^ = correlation coefficient.

**Table 5 plants-12-02174-t005:** Verification result.

Experiment Repetition	Flavonoid Yield (%)	Average Flavonoid Yield (%)	RSD (%)
1st	7.04	7.08	1.58
2nd	7.14		
3th	7.06		

RSD = relative standard deviation.

**Table 6 plants-12-02174-t006:** Single factor experiment design.

Experiments	Solvent Type	Solvent Concentration (%)	Extraction Time (s)	Liquid−Material Ratio (mL/g)
1	Water, ethanol, methanol	80	60	40
2	Ethanol	40, 50, 60, 70, 80	60	40
3	Ethanol	80	40, 50, 60, 70, 80	40
4	Ethanol	80	70	30, 40, 50, 60, 70

## Data Availability

The datasets used and/or analyzed during this study are available from the corresponding author upon reasonable request.

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
