# Peer review of "Improving Flavonoid Accumulation of Bioreactor-Cultured Adventitious Roots in Oplopanax elatus Using Yeast Extract"

_plants, 2023, doi:10.3390/plants12112174_

Round 1
Reviewer 1 Report
The paper entitled ‘Improving Flavonoid Accumulation of Fed-Batch Bioreactor Cultured Adventitious Roots in Oplopanax elatus by Yeast Extract Elicitation’ is a continuation of research on root culture optimisation in bioreactors of this species. Additionally, an extraction process of the flavonoid reach fraction was described.
After reading the text, I have a few comments and suggestions.
Major comments:
1. I have a question, it is related to the design of the study. According to the materials and methods, the results described in Sections 2.1 to 2.3 were performed in the flask. The authors took biomass and medium from the bioreactor, split it into 20 g portions + 100 ml of medium and established agitated cultures to optimise the elicitation process. This should be clearly highlighted in the results and discussed in the Discussion section. The splitting and agitation of the biomass are additional factors that affect the course of the experiment. Agitation is a physical elicitation. It is not known whether the mass ratio of roots to elicitor concentration and medium volume is the same as in the bioreactor. The authors do not write anything about it in the methodology. This should also be stated in the figure captions. At the moment, it follows from the descriptions that these are bioreactor cultures. The descriptions of the graph axes in Figure 2 should be changed. This is the age of the cultures, not the time of the elicitor treatment. Elicitation time was 8 days. All this confuses the reader.
Minor comments:
1. In the first line of the Introduction, there is a spelling mistake in the species name. The authors also provide the name of the family that does not exist. It should be Araliaceae. The paper should be carefully revised to avoid such mistakes.
2. Why do the authors focus on flavonoids? and no other more specific compounds. There is no justification for this. I was unable to find this information in previous works. It should be justified.
3. Why are flavonoids determined on HPLC called monomers? Most flavonoids are monomers.
4. Why were these flavonoids chosen for quantification? Justification in the literature is essential. Why kaempferide? and not the more popular kaempferol? Are there other peaks, other metabolites, flavonoids, or not? An example of a chromatogram could be helpful.
5. Statistical methods (subchapter 4.10) should be carefully rewritten. For example, line 467 ‘A probability of p < 0.05 was considered significant’; lines 121, 138 ‘p < 0.01 compared with control’. There is no information on the analyses used in optimisation of the flash extraction process.
6. Line 295. Citation 32. First names are used instead of surnames. It should be cited as follows: Kochan, E.; Szymczyk, P.; Kuźma, Ł.; Lipert, A.; Szymańska, G. Yeast Extract Stimulates Ginsenoside Production in Hairy Root Cultures of American Ginseng Cultivated in Shake Flasks and Nutrient Sprinkle Bioreactors. Molecules 2017, 22, 880. https://doi.org/10.3390/molecules22060880. The authors should verify all citations to avoid errors.
Reviewer 2 Report
Authors performed intersting study to evaluate conditions of a 5L bioreactor and 250 ml flask culture of Oplopanax elatus, to find conditions of highest flavonoid yield. Authors used YE extract eliciation to improve flavonoid productivity. Moreover, the conditions of flash flavonoid extraction were studied and optimized by RSM method. Article is well planned and performed. Following minor comments sholud be addressed before the publication.
Correct the typographical terror in line 30; should be Oplopanax not
Opolopanax.
Rewrite entences:
Adventitious roots (ARs) 35 of plants could be mass and rapidly produced using plant tissue culture technology [2], 36 thereby considering that AR culture is an efficient way for obtaining raw materials of rare 37 medicinal plants [3].
For new developed plant materials, optimization of extraction process is an important step for the production of products using the materials. Flash extraction was firstly proposed in 1993 by Liu et al. [20], who indicated that unlike traditional extraction methods, flash extraction can complete extraction in a few minutes by grinding and grinding the material into fine particles, which has many advantages, such as sufficient extraction and saving time, solvent and energy.
Section 2.1
Figure 1a: 0Y-axix should be better described. Was the dry weight measured in grams per liter?
Authors should clearly state in text that bioreactor culture was used just to produce AR marterial that was then used in 250 ml flask to perform studies on YE concentration
In the description to Fig. 1 Authors should better describe that AR obtained after initial culture in 5L bioreactors were cultured in 250 mL flasks with different concentrations of YE as descrided in material and methods.
Section 2.2
Authors should clearly state in text that bioreactor culture was used just to produce AR marterial that was then used in 250 ml flask to perform studies on 100 mg/L YE concentration. Now it looks as the YE experiments were performed in 5 L bioreactors not 250 mL flasks.
Figure 2a: 0Y-axix should be better described. Was the dry weight measured in grams per liter?
Section 2.3
Authors should clearly state in text that bioreactor culture was used just to produce AR marterial that was then used in 250 ml flask to perform studies on 100 mg/L YE concentration in 35-days old biorecator culture for subsequent 10 days.
Figure 3a: 0Y-axix should be better described. Was the dry weight measured in grams per liter?
Material and methods
Line 292- state that the experiment lasted for 10 days.
Line 410- correct the citation , should be Kochan et al. Not Ewa et al.
References:
Correct the citation nr 32 to a proper citation format:
MDPI and ACS Style
Kochan, E.; Szymczyk, P.; Kuźma, Ł.; Lipert, A.; Szymańska, G. Yeast Extract Stimulates Ginsenoside Production in Hairy Root Cultures of American Ginseng Cultivated in Shake Flasks and Nutrient Sprinkle Bioreactors. Molecules 2017, 22, 880. https://doi.org/10.3390/molecules22060880
AMA Style
Kochan E, Szymczyk P, Kuźma Ł, Lipert A, Szymańska G. Yeast Extract Stimulates Ginsenoside Production in Hairy Root Cultures of American Ginseng Cultivated in Shake Flasks and Nutrient Sprinkle Bioreactors. Molecules. 2017; 22(6):880. https://doi.org/10.3390/molecules22060880
Chicago/Turabian Style
Kochan, Ewa, Piotr Szymczyk, Łukasz Kuźma, Anna Lipert, and Grażyna Szymańska. 2017. "Yeast Extract Stimulates Ginsenoside Production in Hairy Root Cultures of American Ginseng Cultivated in Shake Flasks and Nutrient Sprinkle Bioreactors" Molecules 22, no. 6: 880. https://doi.org/10.3390/molecules22060880
Minor editing of English language required.
Reviewer 3 Report
The manuscript: “Improving Flavonoid Accumulation of Fed-Batch Bioreactor Cultured Adventitious Roots in Oplopanax elatus by Yeast Extract Elicitation” present results of elicitation of in vitro culture of Oplopanax elatus with yeast extract and is a continuation of research presented by the authors in earlier articles.
Line 47. The term “regulating culture medium” is unclear. The authors should write clearly what they mean.
Line 94. Rather “exceeded” than “reached”.
Line 103, 111, 116. The phrase “YE time” is not correct. The time of exposure of the yeast extract to the plant material was one that is 8 days. The age of the culture was different.
Line 269-271. The phrase “this method could overcome medium inhibition in early culture stage and replenish nutrients “medium inhibition” is not clear. A method could replenish nutrients?
The authors mentioned in the Introduction the studies on the influence of methyl jasmonate on flavonoid production. In the Discussion they should discuss currently presented results in the manuscript with these about MeJa. Moreover, the authors have just published an article: Shuo Yu, Xiao-Han Wu, Miao Wang, Liang-Liang Liu, Wei-Qi Ye, Mei-Yu Jin, Xuan-Chun Piao, Mei-Lan Lian, Optimizing elicitation strategy of salicylic acid for flavonoid and phenolic production of fed-batch cultured Oplopanax elatus adventitious roots, Journal of Biotechnology, 368, 2023, 1-11, which is based probably on the same material and methods. There they studied influence of salicylic acid. In the manuscript they should mention about the results presented in that article and discuss them.
Line 348. There is no need to write the separate chapter describing chemicals. It would be better to write their origin, purity in the brackets when a chemical is mention for the first time.
Which the moment the age of the culture (35-, 40-, 45-days) was measured from? Which moment is start time? Please clarify it in Materials and methods.
How the yeast extract was applied to the medium? As a powder? Sterile?
Were there any replications (flasks) of studied objects? How many?
In fact Conclusions are not conclusions but summary because the authors present shortly results. Please, write this chapter once again. Try to draw conclusions that is why the best age of the culture was just 35 days, why 100 mg/l of YE is the best (not lower nor higher), what is the relationship between flavonoid content and DPPH values, and so on.
There is “radio” a few times in the text instead of “ratio”.
Reviewer 4 Report
The manuscript describes an interesting subject of improving flavonoid production in plants. I have found only one mistake; descriptions of Y axis are not clear and must be improved in the Fig. 1, Fig.2, Fig.3 and Fig.4 (which is another Fig 2).

Round 2
Reviewer 1 Report
The paper has been corrected.